# Patterns of Emergency Room Visits for Respiratory Diseases in New York State in Relation to Air Pollution, Poverty and Smoking

**DOI:** 10.3390/ijerph20043267

**Published:** 2023-02-13

**Authors:** Najm Alsadat Madani, David O. Carpenter

**Affiliations:** 1Department of Environmental Health Science, School of Public Health, 1 University Place, University at Albany, Rensselaer, NY 12144, USA; 2Institute for Health and the Environment, 5 University Place, University at Albany, Rensselaer, NY 12144, USA

**Keywords:** air pollution, poverty, smoking, New York State (NYS), respiratory diseases, asthma, chronic obstructive pulmonary disease (COPD)

## Abstract

We have explored differences in rates of emergency room (ER) visits for respiratory diseases in the counties of New York State (NYS) in relation to levels of air pollution, poverty, and smoking. Air pollution information was derived from the National Emissions Inventory, which provides information on road, non-road, point, and non-point sources of 12 different air pollutants. This information is only available at the county level. Four types of respiratory diseases were considered: asthma, chronic obstructive pulmonary disease (COPD), acute lower respiratory diseases, and acute upper respiratory diseases. Asthma ER visits were elevated in counties with greater total air pollution. All forms of respiratory diseases were elevated in counties with a greater rate of poverty, although this may reflect the fact that poor people often use ERs for routine care. There was a very strong association between rates of smoking for COPD and acute lower respiratory diseases. There was an apparent negative association between smoking and asthma ER visits, but this must reflect the fact that smoking was much more common in upstate counties while asthma was more common in the New York City area, where air pollution is high. Air pollution was much greater in urban than in rural areas. Our evidence indicates that air pollution is the greatest risk factor for asthma attacks, whereas smoking is the greatest risk factor for chronic obstructive pulmonary disease (COPD) and lower respiratory disease. Poor people are more vulnerable to all forms of respiratory diseases.

## 1. Introduction

Respiratory diseases are a major cause of human morbidity and mortality all over the world. In 2015, lower respiratory diseases were the 3rd highest cause of years of life lost, while COPD was the 10th and asthma the 37th in terms of global burden of disease [1]. There are major public health consequences associated with asthma as a well-known respiratory disease, including high morbidity and mortality in severe cases for both children and adults [2]. Around 334 million people worldwide suffer from asthma [3]. Asthma affects 8.4% of Americans, compared with 4.3% of the worldwide population, and its prevalence is on the rise both domestically and internationally. In minorities and lower socioeconomic groups, the impact is most noticeable [4].

Globally, chronic respiratory diseases remain a major cause of death and disability. Chronic respiratory diseases affected over 544 million people worldwide in 2017, an increase of 39.8% since 1990 [5]. The world over, chronic obstructive pulmonary disease (COPD) is a common respiratory disease associated with high morbidity and mortality rates [6]. A number of factors will likely influence the future development of COPD, including longer life expectancy, occupational and environmental exposures, and the increasing prevalence of asthma. Other factors are expected to continue to influence future trends globally, including air pollution and chronic respiratory infections such as tuberculosis [7].

Studies show that air pollution can cause serious health consequences. In urban areas, especially in developing nations, air pollution is a major concern due to its harmful effects on respiratory health [8]. A study examined the association between respiratory emergency department (ED) visits (2011–2015) and biorefinery exposure in residences. A significant increase in respiratory ED visits was reported among residents living within 10 km of biorefineries [9].The causes of respiratory diseases are multiple. Many such diseases are infectious, while others are due to exposure to air pollution or exposure to toxic chemicals. Rates of smoking, poverty and other personal behavioral patterns within different populations are major factors that influence geographic rate differences. While these disparities are particularly pronounced globally, there are also distinct differences in patterns of respiratory diseases within smaller geographic areas.

We have examined the patterns for emergency room visits for respiratory diseases other than cancer within the counties in New York State (NYS). There are 62 counties in New York, and they vary widely in size and population density. They also differ greatly in rates of air pollution, poverty, and smoking. While it would be far preferable to be able to analyze the patterns of respiratory diseases at a smaller geographic unit, data on total levels of air pollution on such a scale as zip codes or census tracts are not available. We have utilized data from the National Emissions Inventory (NEI), which provide results of road, non-road, point source, and non-point source air pollution. Only point source air levels are available at the zip code level, and point source emissions are less than 5% of total air pollution. Therefore, even though counties are relatively large and there will be diverse sources of air pollution within each county, this is the only geographic level for which complete data is available. We have chosen to study asthma, COPD, acute lower respiratory diseases, and acute upper respiratory infections for which patients commonly go to emergency rooms.

### 1.1. Asthma

Asthma is a chronic disease caused, aggravated, or exacerbated by various chemical, physical, and biological exposures both indoors and outdoors [10]. The prevalence of asthma is a consequence of several factors, including genetics, allergies, and environmental exposures. Among environmental risk factors are allergies to dust mites and cockroaches, tobacco, air pollution, microbes, and stress [11]. According to the hygiene hypothesis, early life excessive cleanliness can lead to an immune system that overreacts to triggers [12]. There is a large body of evidence showing that various air pollutants are associated with an increased risk of asthma attacks. Even at levels below the current limit values, long-term exposure to air pollution, specifically from fossil fuel combustion, is associated with adult-onset asthma. A study with a mean follow-up period of 16.6 years for 98,326 participants in three cohorts in Denmark and Sweden revealed that 1965 participants had asthma and reported the associations in fully adjusted models with hazard ratios for particulate matter (PM_2.5_), nitrogen dioxide (NO_2_), and black carbon (BC) [13]. Another study of 4,154,887 respiratory admissions in Italy between 2006 and 2015 showed that short-term exposure to particulate matter (PM) harms the respiratory system, especially for asthma and COPD [14].

### 1.2. Chronic Obstructive Pulmonary Disease (COPD)

COPD has been a major public health problem for decades, and there are numerous studies that show air pollution increases the risk of COPD. Over 3 million people die worldwide every year from COPD [15]. COPD is a group of severe lung diseases such as chronic bronchitis and emphysema that cause difficulty breathing by preventing air from flowing through the airways. The most common cause of COPD in developed countries is smoking [16]. It has been suggested that outdoor air pollution exposure is associated with COPD prevalence and incidence. COPD patients experience increased respiratory symptoms when exposed to outdoor air pollutants [17]. Studies have shown that short-term exposure to air pollution increases the risk of emergency room (ER) visits for lower respiratory diseases in environments with relatively low air pollutants. In one study, the association between ER visits and ambient air pollutants for COPD and acute lower respiratory diseases was tested using a case-crossover design. NO_2_, PM_2.5_, and sulfur dioxide (SO_2_) all showed positive results in males, while ozone (O_3_) and SO_2_ showed positive results in females [18]. A study using data from the UK Biobank on 303,887 individuals aged 40–69 years found that ambient air pollution results in reduced lung function and an increased prevalence of COPD. Moreover, PM_2.5_, particulate matter (PM_10_), and NO_2_ concentrations are all associated with a higher prevalence of COPD [19].

### 1.3. Acute Lower Respiratory Diseases

In this paper, we studied acute bronchitis and bronchiolitis as acute lower respiratory diseases. Several studies have found that exposure to gaseous air pollution increases the risk of acute lower respiratory diseases. A study on children who visited ER services for bronchiolitis during the period 1997–2001 in Paris suggests that air pollution may contribute to acute severe bronchiolitis. Air pollutants such as PM_10_, black smoke, SO_2,_ and NO_2_ were positively associated with hospitalizations and consultations after adjusting for public holidays, holidays, and meteorological variables [20]. Another study on 29,688 children’s hospital admissions in Hong Kong from 2008 to 2017 revealed that elevated NO_2_ concentration was associated with higher hospitalization rates for acute bronchiolitis; the risk of hospitalization for acute bronchiolitis increased statistically significantly with cumulative NO_2_ exposure over the range 66.2–119.6 µg/m^3^ [21].

### 1.4. Acute Upper Respiratory Diseases

This paper examines acute upper respiratory diseases including acute laryngopharyngitis, acute upper respiratory infections of other multiple sites, and acute upper respiratory infections of unspecified sites. Epidemiological studies have shown an association between exposure to air pollution and an increased risk of upper respiratory diseases. A study on ER visits data from three hospitals in Lanzhou, China between January 2014 and December 2018 revealed that an increase in ER visits for upper respiratory tract infections was linked to short-term exposure to air pollution, particularly SO_2_ and carbon monoxide (CO). At lag 07, the percentage increase for each 1mg/m^3^ increase in CO was 20.8% (95% CI: 11.8, 30.5) [22]. According to another study on pediatric ER visits in 66 hospitals in Shanghai, China from 2016 to 2018, short-term exposure to air pollutants such as ultrafine particles could increase the risk of ER visits for upper respiratory tract infections among children (1.14, 95% CI: 1.02–1.28) [23]. Several studies indicate that ambient air pollution increases the risk of upper respiratory tract infections. According to a study conducted at Wuhan University Hospital in Wuhan, China, gaseous pollutants negatively impact health more than solid pollutants. A significant number of clinic visits for upper respiratory tract infections were caused by SO_2_ and NO_2_, to which women seemed more susceptible than men [24]. Based on data on hospital visits from 1 October 2010 to 30 September 2012 from the Beijing Medical Claim data, researchers found that a 10 µg/m^3^ increase in PM_2.5_ concentration was associated with 0.84% (95% confidence interval, 0.05–1.64%) increases in hospital admissions for upper respiratory tract infections at lag 0–3 days, but not with emergency rooms or outpatients [25].

While respiratory infections are clearly due to infectious agents, the vulnerability to infection varies with factors such as poverty, immune system status, and access to and quality of health care. In this paper, we seek to clarify the relationships between air pollution, smoking, and poverty with respiratory diseases, to gain a better understanding of the importance of the role these factors have in human health.

## 2. Methods

### 2.1. Health Data

Hospitals located in and regulated by NYS are required to report all diagnoses, up to 15 per inpatient or ER visit, to the NYS Department of Health (NYS DOH) upon discharge. This diagnostic report system is known as the Statewide Planning and Research Cooperative System (SPARCS), and was established in 1979. The SPARCS data that we have contains all ER visits to state-regulated ERs (all but Veterans Administration or Indian Health Service clinics) from 2010–2018. With the help of R studio software version 2022.07.1 Build 554, the total number of ER visits and primary diagnoses of respiratory diseases for each year was calculated.

The International Classification of Diseases, Ninth Revision, and Clinical Modification (ICD-9-CM) and The International Classification of Diseases, Tenth Revision, and Clinical Modification (ICD-10-CM,) and the SPARCS Data Dictionary were used to find the appropriate code and category for each respiratory disease. Codes are as follows: asthma (ICD9 = 493; ICD10 = J45), COPD (ICD9 = 490, 491, 492, 494, 496; ICD10 = J40, J41, J42, J43, J44, J47), acute lower respiratory infections (acute bronchitis and bronchiolitis) (ICD9 = 466; ICD10 = J20, J21), and acute upper respiratory infections of multiple or unspecified sites (ICD9 = 465; ICD10 = J06).

The number of ER visits for each disease for each year was calculated according to the ICD code. In the next step, this data was categorized based on each county to have the crude number of ER visits for each disease per year at the county level. The Federal Information Processing Standard (FIPS) code was used to categorize data at the county level. In the last step, we calculated the rate of ER visits for each county base on the crude number of ER visits and the population of that county.

### 2.2. Population Data

The total population in each county of NYS was downloaded from the United States Census Bureau database for 2010 and the estimated population for 2018 was downloaded from NYS DOH. We interpolate the population numbers between 2010 and 2018 using the Equation (1) below.
(1)$$\begin{array}{l} \mathrm{Population}\;\left(\mathrm{Year}\right)\\ \;=\mathrm{Population}\;2010+\left(\mathrm{Year}\;-2010\right)*(\mathrm{Population}\;2018\\ \;-\;\mathrm{Population}\;2010)/\left(8\right)\;\mathrm{where}\;\mathrm{year}\;=2011,\;2012,\ldots ,2017 \end{array}$$

The rates of ER visits for respiratory diseases were calculated by dividing the crude number of ER visits for each respiratory disease in each county by the population of that county. The total rate of ER visits from 2010–2018 was calculated by adding the rate of ER visits for each respiratory disease in the years 2010–2018. The rate of ER visits for each respiratory disease by county is given by the following Equation (2).
(2)$$\begin{array}{l} \mathrm{The}\;\mathrm{rate}\;\mathrm{of}\;\mathrm{ER}\;\mathrm{visits}\;\mathrm{for}\;\mathrm{each}\;\mathrm{respiratory}\;\mathrm{disease}\;\mathrm{by}\;\mathrm{county}\;\\ \;=\left(\mathrm{the}\;\mathrm{number}\;\mathrm{of}\;\mathrm{ER}\;\mathrm{visits}\;\mathrm{for}\;\mathrm{each}\;\mathrm{respiratory}\;\mathrm{disease}\right)/\left(\mathrm{county}\;\mathrm{population}\right) \end{array}$$

### 2.3. Exposure Data

The exposure data came from the United States Environmental Protection Agency (US EPA) National Emissions Inventory (NEI), a data set of air emissions and pollutants from point, non-point, road, and non-road sources by county. The FIPS code was used to merge data for each pollutant at the county level from all four sources of pollution. The total pollution was calculated by adding together the data for each county of NYS for 2011, 2014, and 2017 divided by 3 for each chemical—total volatile organic compounds (VOCs), PM_2.5_, acetaldehyde, formaldehyde, benzene, ethylbenzene, toluene, xylene, SO_2_, NO_2_, CO, and lead—from all four different sources of pollution (point, non-point, road, and non-road). Total pollution for lead was only available for point and non-point sources. Land area in square miles for each county was found in the NYS DOH database. Pollution per square mile was calculated by dividing the total pollution for each chemical by the land area in square miles.

Point sources include estimates of emissions from large, fixed, stationary sources such as industrial facilities, electric power plants, airports, as well as smaller industrial, commercial, and non-industrial sources. In addition to point sources, non-point sources include sources that are too small in magnitude to be reported as point sources, such as household heating, commercial combustion, asphalt paving, and commercial and consumer solvent use. Emissions from on-road vehicles running on gasoline, diesel, or other fuels are included in the NEI on-road sources. Light-duty and heavy-duty vehicles operating on roads, highway ramps, and while idling are also considered on-road sources. Fuel sources that use gasoline, diesel, and other fuels are included among NEI non-road sources. There are several types of non-road sources, including construction equipment, lawn, and garden equipment, aircraft ground support equipment, locomotives, and commercial marine vessels.

With the help of ArcGIS software version Desktop 10.8.1, maps were created to show the total air pollution (other than lead) per square mile. The shape files were downloaded from the Topologically Integrated Geographic Encoding and Referencing system (TIGER) that represents the U.S. Census Bureau’s geographic spatial data.

### 2.4. Poverty Data

Each county’s percentage of poverty was downloaded from the United States Census Bureau database for 2010 to 2018 separately. The data is called the Small Area Income and Poverty Estimates (SAIPE) Program which gives the percentage of people in poverty. The SAIPE program estimates are based on several data sources, such as aggregate tax, Supplemental Nutrition Assistance Program (SNAP) benefits, and poverty incidence data. Some of these data sources are publicly available, while others are not. In SAIPE estimates, poverty incidence is defined similarly to that in the American Community Survey (ACS). Depending on family size and composition, the Census Bureau determines who is poor based on money income thresholds. Poverty is defined when a family’s total income is less than the threshold for a family. In this paper, we calculated the total percentage of poverty by adding the data together for each county in NYS from 2010 to 2018 and dividing by 9, to obtain annual rates.

### 2.5. Smoking Data

The crude rate of smoking in each county was downloaded from The Behavioral Risk Factor Surveillance System (BRFSS) database for 2016 and 2018. The total rate of smoking was determined by adding the crude rate of smoking in 2016 and 2018 together for each county divided by 2. We defined different statistical models to determine the impact of smoking on the rate of ER visits for respiratory diseases.

### 2.6. Statistical Analysis

R studio software version 2022.07.1 Build 554 was used to create figures. Correlation analysis was used to assess the relationships between 12 different air pollutants and quantify how closely related any two variables are. A positive correlation occurs when air pollutants change in the same direction. The FIPS code was used to merge health data and pollution data to have the rate of ER visits for respiratory diseases and pollution per square mile at the county level in one dataset.

Regression analysis aims to determine how the rate of ER visits for respiratory diseases is impacted by exposure to 12 air pollutants (total VOCs, PM_2.5_, acetaldehyde, formaldehyde, benzene, ethylbenzene, toluene, xylene, SO_2_, NO_2_, CO, and lead). All these chemicals were aggregated as a separate variable, and regression models were developed to assess the association between air pollution exposure and the rate of ER visits for respiratory diseases.

In order to determine the best fitting model for air pollution adjusted for smoking and poverty, we compared different family regression models. For the purpose of assessing the quality of regression models and choosing the best fitting model, we used a performance package. To measure the performance of a regression model, different statistical regression metrics were calculated, including root mean squared error (RMSE) and residual standard error (RSE), or sigma. The regression models were conducted with *p* < 0.05 as the threshold for statistical significance. After choosing the best family model to be quasi-Poisson regression, we ran different models in our study to assess the association between air pollution and ER visits for respiratory diseases, and to determine the impact of smoking and poverty on the rate of ER visits for respiratory diseases.

Among unadjusted models, our response variable is the rate of ER visits; the predictor variable in the first group of models is air pollution, in the second group of models it is poverty, and in the third group of models it is smoking.

Among adjusted models, the first group of multiple regression models assesses the association between exposure to air chemicals and the rate of ER visits adjusted for poverty. The second group of multiple regression models examines the association between air pollution and the rate of ER visits adjusted for smoking, and the third group of multiple regression analyses was run to find the relationship between air pollutants and respiratory diseases adjusted for both poverty and smoking.

In the last step, models adjusted for smoking and poverty were compared to unadjusted models using the performance package. Different statistical regression metrics including Nagelkerke R^2^, RMSE, and RSE (known as sigma), were calculated to compare these different regression models.

## 3. Results

### 3.1. Rate of ER Visits for Diseases

Figure 1 shows the ER visits rate for respiratory diseases, and Appendix A presents details. Figure 1a presents the rate of ER visits for asthma. The highest rates of ER visits for asthma were in Bronx, Kings (Brooklyn), and New York (Manhattan) Counties. Asthma ER visits were also elevated in the New York City suburbs, Monroe County which contains the city of Rochester, the Capital Region of Albany and Schenectady Counties, and Montgomery County which contains the city of Amsterdam. The counties with the least ER visits for asthma were Saratoga, Schoharie, and Tioga, which are rural. Figure 1b,c shows the ER visits rate for COPD and acute lower respiratory diseases. In contrast to the situation with asthma, the New York City counties and the surrounding suburban areas all showed low rates of acute lower respiratory diseases and COPD. These diseases were also relatively low in the upstate urban areas but were very elevated in rural counties, especially those in the northern part of the state. Figure 1d presents the ER visits rate for acute upper respiratory diseases. Montgomery, Cortland, and Bronx are the counties with the highest rate of ER visits for upper respiratory diseases, and Greene, Saratoga, and Tioga are counties with the lowest rate.

The National Center for Health Statistics provides information about urban—rural classification of counties for 2013. According to this dataset Allegany, Chenango, Delaware, Essex, Greene, Hamilton, Lewis, Schuyler, Sullivan, and Wyoming are among rural counties. Urban counties include Bronx, Erie, Kings, Monroe, New York, Queens, and Richmond County. Bronx, Kings, New York, Queens and Richmond Counties are the five NYC counties; Erie County contains the city of Buffalo; and Monroe contains the city of Rochester.

### 3.2. County-Level Air Pollution

The total air pollution per square mile at the county level was obtained from the National Emissions Inventory and is shown in Figure 2a and in Appendix A. The counties with the highest rate of air pollution per square mile are New York, Kings, and Queens—all of which are New York City counties. Air pollution is also high in Nassau County on Long Island, as well as the other New York City suburbs. There are clear elevations in air pollution in the major upstate urban areas: Erie and Niagara Counties in the west, where Buffalo and Niagara Falls are located; Monroe County to the east, home to Rochester; further east in Onondaga County with the city of Syracuse; and in the Albany capital district still further east. This is clear evidence that traffic is a major contributor to local air pollution. Counties with the lowest air pollution per square mile are Hamilton, Delaware, and Lewis—all rural counties. The pollution data for four of the counties studied are not available in the NEI-EPA database, distinguishable by the gray color on the map. Also shown in Figure 2b is a table giving the correlations among the 12 air pollutants that are included in the total air pollutants values. All components are very highly correlated, which prevents us from examining the health effects of individual chemicals. The least highly correlated air pollutant is SO_2_, which comes from coal combustion, but even SO_2_ has 95% correlation with most of the others.

### 3.3. Poverty and Smoking

Figure 3a shows the poverty rate in NYS; the top three counties with the highest poverty rate are Bronx, Kings (Brooklyn), and Chautauqua in the low western part of the state. The counties with the lowest rate of poverty are Saratoga, Nassau, Putnam, which are relatively affluent suburban counties. Figure 3b shows the county-wide pattern of smoking. Smoking rates are much higher in rural than urban counties, and this is true even in urban counties with a relatively high rate of poverty, such as the Bronx. Orleans, Oswego, Franklin, and all rural counties in northern New York are counties with the highest rate of smoking. Nassau, Westchester, Rockland, and all New York City suburbs are counties with the lowest rate of smoking in NYS. Appendix A summarizes the data about the rate of poverty and smoking at the county level.

### 3.4. Regression Models

We moved from one regression family to another family to find the best fitting model for our study. Different statistical regression metrics such as RMSE and RSE (or sigma) were calculated to measure the performance of a regression model. Table 1 summarizes the results of the models’ comparison. By comparing RMSE with the sigma of the different regression family models, we could see that the quasi-Poisson model is the best model to define the association between exposure to air pollution and the rate of ER visits adjusted for smoking and poverty.

#### 3.4.1. Air Pollution Models

Table 2 summarizes the association between exposure to air pollution and the rate of ER visits for respiratory diseases, and Figure 4 represents this association. According to the results of quasi-Poisson regression models, air pollution is positively associated with a statistically significant increased rate of ER visits for asthma (*p*-value = 7.14 × 10^−5^) but not for the other respiratory diseases. The rates of ER visits for acute lower respiratory diseases and COPD are negatively associated with air pollution. Acute upper respiratory diseases have a *p*-value of 0.176, which is not an indicator of significant association. It is apparent that the data from only a few counties strongly influence these results.

#### 3.4.2. Poverty Models

Air pollution is not the only possible risk factor for an increased rate of ER visits for different diseases. Other expected risk factors include poverty and smoking. Figure 5 represents the association between poverty and respiratory diseases. As shown in Table 3, poverty is associated with elevated risk for all these diseases, and the associations are statistically significant. For COPD, the *p*-value is 0.064. Appendix A presents the model results of the association between exposure to air chemicals and the rate of ER visits adjusted for poverty. The findings of these multiple regression models confirm previous findings that poverty is highly associated with all respiratory diseases. 

#### 3.4.3. Smoking Models

The association between smoking and diseases is shown in Figure 6 and Table 4. Smoking is highly associated with COPD (*p*-value = 0.000487) and acute lower respiratory diseases (*p*-value = 0.012402), and those associations are statistically highly significant. For asthma, the *p*-value is 0.012723 and the association is negative. As discussed below, this finding cannot be correct. For acute upper respiratory diseases, the *p*-value is 0.966042 which is not an indicator of a statistically significant association. Exposure to air chemicals and the rate of emergency room visits adjusted for smoking are presented in Appendix A. Across all diseases, smoking is associated with high rates of COPD and acute lower respiratory disease.

#### 3.4.4. Air Pollution Models Adjusted for Poverty and Smoking

Table 5 provides us with the results of the multiple regression models to determine the association between air pollution and respiratory diseases adjusted for both smoking and poverty. Poverty is associated with all respiratory diseases. Poverty associations are statistically highly significant for all respiratory diseases. Smoking is positively associated with COPD and negatively with asthma. Among all diseases adjusted for both smoking and poverty, air pollution is statistically highly linked to asthma and negatively related to COPD.

### 3.5. Comparison of the Adjusted Models with the Unadjusted Model

Models adjusted for smoking and poverty were compared to unadjusted models to find the best fitting model to define this association. As in previous sections, different statistical regression metrics such as Nagelkerke R^2^ root mean squared error (RMSE), and residual standard error (RSE—known as sigma)—were applied to measure the performance of the regression models. Table 6 summarizes the results of the models’ comparison. By comparing Nagelkerke R^2^, RMSE, and sigma of the adjusted model to the unadjusted model, we could see that for all diseases, the adjusted model is a better model to define the association between exposure to air pollution and the rate of ER visits.

## 4. Discussion

### 4.1. Respiratory Diseases Are Very Serious Contributors to Global Morbidity and Mortality

In this research, air pollution has been shown to be the greatest risk factor for asthma attacks. A higher level of total air pollution was associated with more asthma ER visits. The results of our research are in line with many studies that confirm the link between exposure to air pollution and the increased rate of asthma. While the results of this current study provide interesting information about the geographic differences in ER visits for respiratory diseases in one US state, it is clear that none of the models is adequate because we have been unable to clearly detect the expected adverse effect of air pollution on any of the diseases studied other than asthma. There are several possible reasons for this. There are likely to be other variables beyond smoking and poverty that influence the results. Age may be one, as residents in many rural communities are older than urban residents and are also much less diverse than in urban areas. We have not controlled for race/ethnicity or sex. County-level data may be just too large a geographic area and the sources of air pollution and the residents too diverse. However, we clearly find the expected associated associations with smoking, especially with COPD, and poverty with all diseases.

### 4.2. Patterns of Respiratory Diseases in NYS: Role of Urban vs. Rural Areas, and of Population Density

The amount of air pollutants released in NYS is by far the greatest in the New York City counties, which is not surprising because of the greater population density here and the very high level of automobile and truck traffic, especially diesel buses and trucks. There are also clear elevations in the other larger urban areas, from the Buffalo/Niagara Falls area in the west, the upstate urban areas in Rochester, Syracuse, and the city area of Albany, Schenectady, and Troy in the East. In urban and rural settings, pollutants vary in level and type. Several factors can affect air pollution in an urban area, including the layout of the city, the road density, the population density, and the localized population concentration [26]. The quality of the air improves with a shift from urban to rural areas. Compared with large central metropolitan counties, noncore counties had fewer mean number of PM_2.5_ days, decreasing from a total of 11.21 to 0.95. Compared with large central metropolitan counties, nonmetropolitan counties had lower mean PM concentrations, decreasing from 11.15 µg/m^3^ to 8.87 µg/m^3^ [27].

The most striking observation of the levels of air pollutants is the great degree to which their concentrations are correlated. As it was shown in Figure 2b, of the 12 air pollutants studied, most had correlations of 0.99 to 1.00. Only two showed less than a 0.95 correlation, one being lead, which isn’t surprising since there are few sources of lead to the environment now. The other is SO_2_, a product of burning high sulfur coal or gas. The tight correlation among the other air pollutants is strong evidence that oil and gas combustion is the common source of formation and release of all of the remaining chemicals, including the other five criteria air pollutants and the various VOCs.

There are two major implications of these results. Because these pollutants are all released together, it is very difficult if not impossible to identify which one or ones are responsible for any particular disease. The second is that the major source is vehicle traffic and fossil fuel combustion. This further indicates how important moving away from fossil fuel combustion is, if we are to both reduce climate change and protect the public from the health hazards resulting from air pollution.

### 4.3. Role of Poverty

Poverty is well known to be one of the strongest risk factors for ill health. This is a consequence of several different factors. Poor people often live in areas with more contamination, have less access to quality health care, and practice less healthy lifestyles. Our data show a large and consistent increase in all respiratory diseases we have monitored in relation to poverty, including both upper and lower respiratory infections. Our results are consistent with other studies and COPD is more common in rural areas. A study in New Hampshire reported that rural counties had higher rates of COPD exacerbation encounters than non-rural counties when poverty and other sociodemographic factors were taken into account [28].

While to a certain degree, respiratory infections may be elevated in people living in poverty because of greater exposure to infectious agents, this elevation may primarily reflect access to care. Poor people often do not have health insurance or access to primary preventive health care. When they are ill, they then go to the local emergency room. We have no way to determine the degree to which this is a factor with the present data, but we consider it likely to be a very important factor.

Another important consideration in this study is that poor people are more likely to use emergency rooms for primary care than is the case for more affluent people. There are many people in the US that do not have health insurance, and for these people the ER is their major source of health care. There are two major federal healthcare programs. Medicaid provides access to health care for poor people and people with some disabilities and Medicare provides access to health care for old people. But others must depend on private insurance, which is expensive. The result is that many working poor and otherwise healthy younger people lack any form of health insurance and use emergency rooms when ill. Thus, the population of individuals found in our SPARCS ER dataset does not reflect a general cross-section of the state population.

### 4.4. Role of Smoking

We had not expected to see such a clear difference in rates of smoking in relation to urban as compared to rural counties. Smoking rates in all of the primary urban counties are not high. This is especially true for NYC and is particularly striking in that smoking rates in the upstate counties are highly correlated with poverty levels, but that is not true in NYC where there is a lot of poverty. Despite high rates of poverty, the smoking rates in NYC and even in the surrounding counties are very low, and are much lower again than in the upstate urban areas.

Our study shows that COPD is highly associated with smoking, and these findings are consistent with other studies. According to Strzelak et al., tobacco smoke causes inflammation, allergies, and other lung diseases by altering immune responses because exposure to tobacco smoke results in an imbalance of antioxidants and oxidants that leads to oxidative stress, inflammation of the mucosa, as well as increased expression of inflammatory cytokines [29].

### 4.5. Strengths

This study provides information on outdoor air pollution from all sources, including road, non-road, point source, and non-point source. Another important strength herein is the availability of a very large number of state-wide ER visits, with data from 2010 to 2018. This gives it strong statistical significance. Taking advantage of its geographical targeting, policy and public health leaders in New York can better understand the prevalence and distribution of respiratory diseases.

### 4.6. Limitations

Other than asthma, we have been unable to document the known effect of outdoor air pollution on respiratory diseases even with the use of multiple statistical models. This must be due to confounders that we have not controlled for or inadequacies of the models we have used. A wide range of respiratory diseases is linked to air pollution, as described in the Introduction. We have no control over indoor air pollution, which is probably more important than outdoor air pollution. We do not have information about people’s occupations and the air pollutants that they might be exposed to in their jobs. The data on ozone is not available in the NEI-EPA data set, and in our analysis of the adverse effect of air pollution, we did not consider ozone. Another limitation is that we find each of the 12 different air pollutants to be so highly correlated at the county level that this prevents us from identifying which specific contaminants are most associated with the different respiratory diseases.

### 4.7. Suggestions for Future Research

Further study of the effects of air pollutants on respiratory health in other states or countries would be helpful to confirm some of the results of this study. It would be interesting to compare findings in order to have a deeper understanding of the adverse effects of air pollution on human health. The model could be improved by adding other confounders such as age, sex, or race. We recommend this study be conducted in a smaller geographical area, such as zip codes, although there the problem of individual air pollutants not being commonly available in smaller geographic areas would remain. There needs to be further research in order to elucidate the contribution of specific VOCs (especially benzene and formaldehyde) and ozone as causes of respiratory diseases.

## 5. Conclusions

We have monitored levels of 12 air pollutants and found that total air pollution is much greater in urban than in rural areas, while rates of smoking are much higher in rural areas. We find a very strong association between geographic differences in levels of 12 air pollutants and emergency room visits for asthma. However, because concentrations of 12 different air pollutants are so tightly coupled, it is not possible to determine which substances contribute the most to the risk of asthma attack. There is also a very strong association between rates of smoking and COPD and acute lower respiratory diseases. Differences in levels of poverty show that poverty increases the rates of all respiratory diseases, although this may at least in part reflect the fact that many poor people use emergency rooms for primary care. However, our data and models are unable to detect the known elevated risk of outdoor air pollution on respiratory diseases other than asthma. Unfortunately, our study has not been able to answer the critical question of the relative roles of particulate matter, nitrogen and sulfur gases, and specific VOCs, on respiratory diseases, because levels of all the different air pollutants are very tightly coupled within relatively large geographic areas such as counties.

## Figures and Tables

**Figure 1 ijerph-20-03267-f001:**
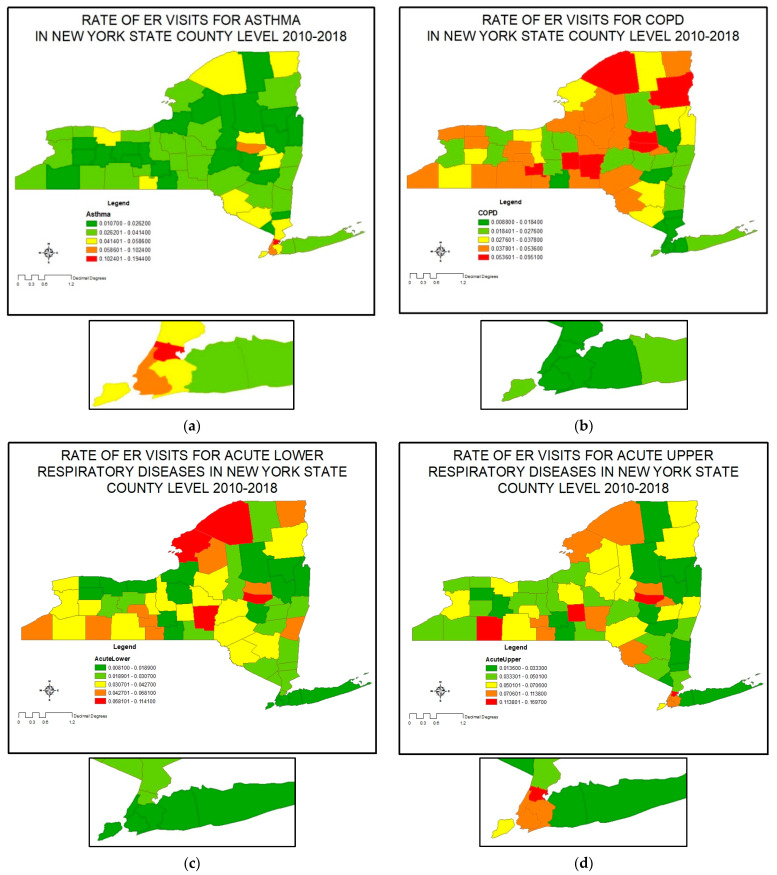
Rate of ER visits for respiratory diseases 2010–2018 by county: (**a**) asthma; (**b**) COPD; (**c**) acute lower respiratory diseases; (**d**) acute upper respiratory diseases. The insert shows the New York City region at higher magnification.

**Figure 2 ijerph-20-03267-f002:**
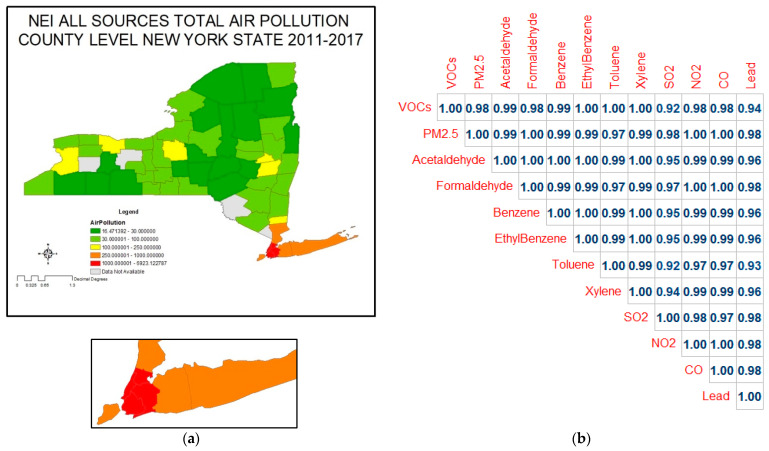
(**a**) Total air pollution per square mile by county; (**b**) correlation between 12 air pollutants.

**Figure 3 ijerph-20-03267-f003:**
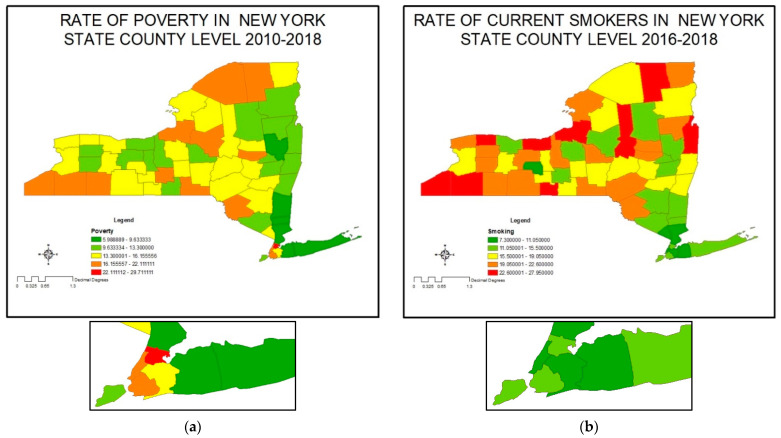
Rate of poverty and smoking in New York State by county: (**a**) poverty; (**b**) smoking.

**Figure 4 ijerph-20-03267-f004:**
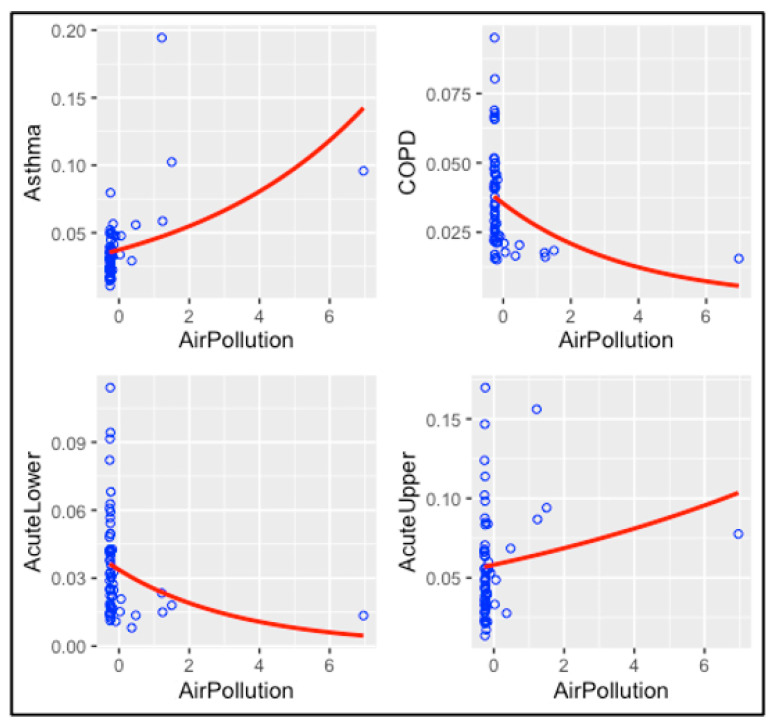
The association between air pollution and respiratory diseases.

**Figure 5 ijerph-20-03267-f005:**
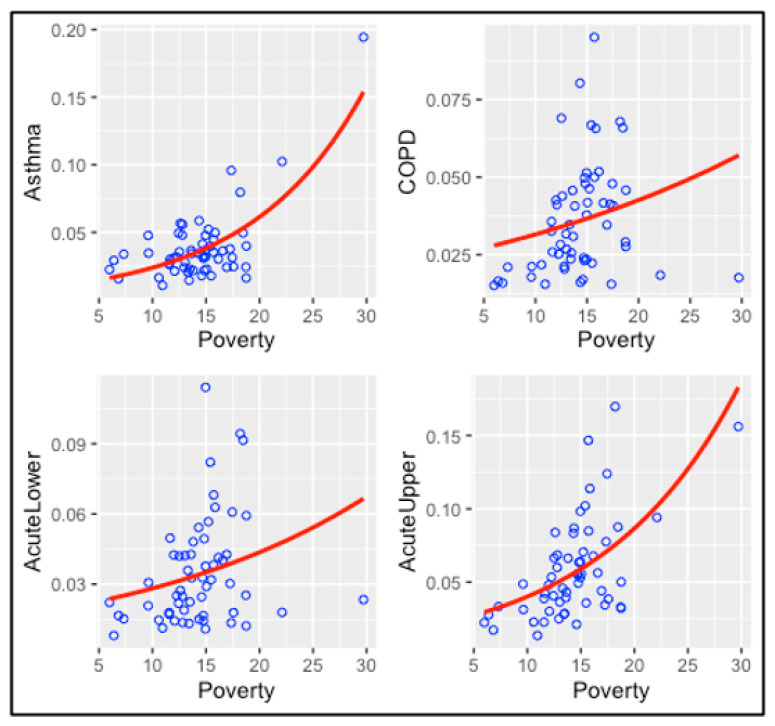
The association between poverty and respiratory diseases.

**Figure 6 ijerph-20-03267-f006:**
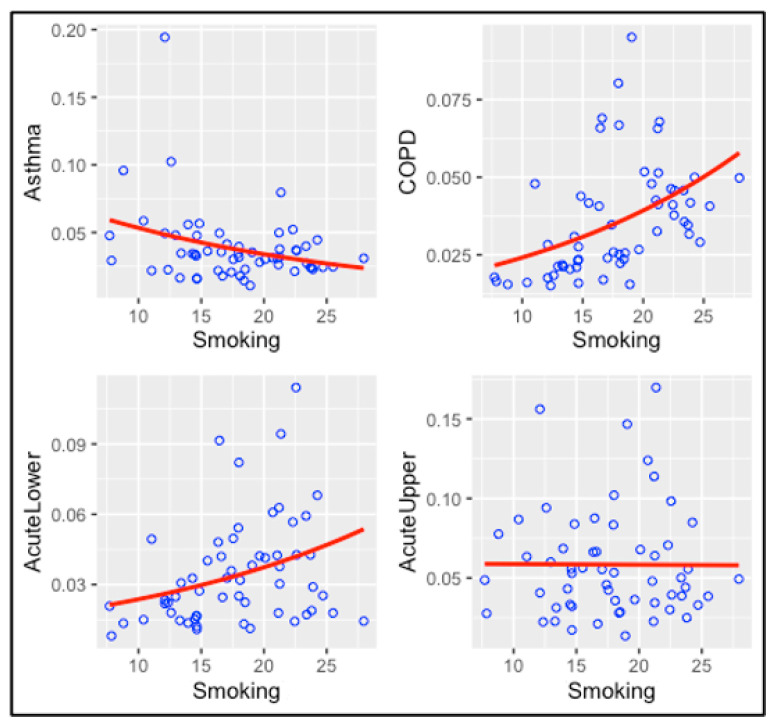
The associations between smoking and respiratory diseases.

**Table 1 ijerph-20-03267-t001:** Comparison of models from the different regression model families.

Respiratory Disease	Name	Model	RMSE	Sigma	Performance Score
Asthma	Quasi-Poisson	glm	0.013	0.068	0.988
	Quasibinomial	glm	0.013	0.070	0.964
	Gaussian	glm	0.017	0.017	0.500
	Linear	lm	0.017	0.017	0.500
	Inverse Gaussian	glm	0.015	2.221	0.250
COPD	Quasi-Poisson	glm	0.015	0.077	0.985
	Quasibinomial	glm	0.015	0.079	0.985
	Gaussian	glm	0.015	0.016	0.763
	Linear	lm	0.015	0.016	0.763
	Inverse Gaussian	glm	0.016	2.094	0.000

**Table 2 ijerph-20-03267-t002:** The association between air pollution and respiratory diseases.

Respiratory Disease	Term	Estimate	Standard Error	Statistic	*p*-Value
Asthma	Intercept	−3.287	0.081	−40.601	3.15 × 10^−43^
	Air Pollution	0.192	0.045	4.290	7.14 × 10^−5^
COPD	Intercept	−3.342	0.064	−51.917	4.70 × 10^−49^
	Air Pollution	−0.263	0.129	−2.047	0.045411
Acute Lower Respiratory	Intercept	−3.391	0.087	−38.936	3.03 × 10^−42^
	Air Pollution	−0.285	0.182	−1.570	0.12211
Acute Upper Respiratory	Intercept	−2.844	0.077	−36.937	5.21 × 10^−41^
	Air Pollution	0.083	0.060	1.372	0.175528

**Table 3 ijerph-20-03267-t003:** The association between poverty and respiratory diseases.

Respiratory Disease	Term	Estimate	Standard Error	Statistic	*p*-Value
Asthma	Intercept	−3.331	0.059	−56.223	5.89 × 10^−51^
	Poverty	0.359	0.045	8.053	6.33 × 10^−11^
COPD	Intercept	−3.327	0.064	−52.220	3.42 × 10^−49^
	Poverty	0.114	0.060	1.888	0.064274
Acute Lower Respiratory	Intercept	−3.381	0.083	−40.603	3.14 × 10^−43^
	Poverty	0.165	0.076	2.172	0.034117
Acute Upper Respiratory	Intercept	−2.886	0.064	−45.429	6.99 × 10^−46^
	Poverty	0.292	0.051	5.671	5.18 × 10^07^

**Table 4 ijerph-20-03267-t004:** The associations between smoking and respiratory diseases.

Respiratory Disease	Term	Estimate	Standard Error	Statistic	*p*-Value
Asthma	Intercept	−3.282	0.085	−38.813	3.60 × 10^−42^
	Smoking	−0.214	0.083	−2.574	0.012723
COPD	Intercept	−3.346	0.062	−54.365	3.74 × 10^−50^
	Smoking	0.229	0.062	3.704	0.000487
Acute Lower Respiratory	Intercept	−3.389	0.082	−41.178	1.46 × 10^−43^
	Smoking	0.214	0.083	2.584	0.012402
Acute Upper Respiratory	Intercept	−2.840	0.078	−36.547	9.21 × 10^−41^
	Smoking	−0.003	0.078	−0.043	0.966042

**Table 5 ijerph-20-03267-t005:** The association between air pollution and respiratory diseases adjusted for smoking and poverty.

Respiratory Disease	Term	Estimate	Standard Error	Statistic	*p*-Value
Asthma	Intercept	−3.347	0.048	−69.898	1.20 × 10^−54^
	Air Pollution	0.077	0.034	2.252	0.028437
	Smoking	−0.174	0.052	−3.361	0.00143
	Poverty	0.316	0.035	9.007	2.42 × 10^−12^
COPD	Intercept	−3.364	0.060	−55.954	1.69 × 10^−49^
	Air Pollution	−0.221	0.131	−1.690	0.096765
	Smoking	0.148	0.069	2.149	0.036128
	Poverty	0.138	0.067	2.072	0.043017
Acute Lower Respiratory	Intercept	−3.429	0.083	−41.396	1.37 × 10^−42^
	Air Pollution	−0.347	0.214	−1.624	0.110298
	Smoking	0.093	0.093	1.002	0.321012
	Poverty	0.229	0.088	2.593	0.012203
Acute Upper Respiratory	Intercept	−2.885	0.065	−44.474	3.17 × 10^−44^
	Air Pollution	0.005	0.065	0.075	0.940876
	Smoking	−0.022	0.071	−0.305	0.761745
	Poverty	0.289	0.055	5.258	2.56 × 10^−6^

**Table 6 ijerph-20-03267-t006:** Comparison of the adjusted models with the unadjusted model.

Respiratory Disease	Model	R^2^ Nagelkerke	RMSE	Sigma
Asthma	Adjusted	0.671	0.013	0.068
	Unadjusted	0.239	0.025	0.102
COPD	Adjusted	0.313	0.015	0.077
	Unadjusted	0.119	0.017	0.086
Acute Lower Respiratory	Adjusted	0.244	0.020	0.102
	Unadjusted	0.079	0.022	0.111
Acute Upper Respiratory	Adjusted	0.351	0.028	0.111
	Unadjusted	0.032	0.034	0.133

## Data Availability

Exposure data: https://www.epa.gov/air-emissions-inventories/national-emissions-inventory-nei Poverty data: https://www.census.gov/data-tools/demo/saipe/#/ Smoking data: https://health.data.ny.gov/Health/Behavioral-Risk-Factor-Surveillance-System-BRFSS-H/jsy7-eb4n Health data: SPARCS dataset is not an open access and needs application to be submitted to New York State, Department of Health. https://www.health.ny.gov/statistics/sparcs/ Population data: https://www.health.ny.gov/statistics/vital_statistics/2018/table02.htm

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
