# Peer review of "Patterns of Emergency Room Visits for Respiratory Diseases in New York State in Relation to Air Pollution, Poverty and Smoking"

_ijerph, 2023, doi:10.3390/ijerph20043267_

Round 1
Reviewer 1 Report
General comments:
This study employed data from National Emissions Inventory and SPARCS to discuss the influence of various factors on people suffering from respiratory diseases in the counties in New York State. The influence of 12 different air pollutants was discussed using correlation analysis. Poverty rate and smoking data were also used to assess the impact from non-emitting sources using family regression models.
While the results of this study are important for understanding the role of different factors in respiratory diseases, some issues in the manuscript design and expression are still evident, as listed below. Therefore, I suggest a major revision before its publication.
Major concerns:
1. Some of the data in the introduction is not clearly attributed, for instance, L31-L32 and those listed below. I suggest the authors carefully read through this manuscript and make sure all literature citations are sufficient and appropriate.
2. Some of the content used to explain the significance of the research has been put into the discussion sector, and I suggest integrating these contents into the introduction sector to make the paper more organized.
3. The authors repeatedly emphasize regional differences in ER visits and indicate that poor people are more likely to go to ER for treatment of their respiratory diseases. As can be seen from Figure 1, ER visit rates in different counties do vary greatly. Is it appropriate to use ER visit data to characterize prevalence rates? The data of respiratory diseases in this paper are completely derived from the data of ER visit, so the reliability of the data of regions with low poverty rate will be low, and the conclusion presented by the author that poverty rate has a great impact on the prevalence of respiratory diseases may not be valid.
4. The authors mentioned that most had correlations of 99 to 100 of the 12 air pollutants studied. However, these data are not explicitly given in the main text or supplement.
About the graphical
5. In Sector 3.1, the authors compared ER visit between urban and rural counties. However, Figure 1 did not indicate urban and rural counties. I suggest adding some simple elements or instructions to make the distribution of urban and rural counties more intuitive.
Main text
6. L32-34 I think the sources of the data should be presented here.
7. L64-66 Suggest putting data from recent years.
8. L75-76 Suggest presenting the sources of the data.
9. Authors should state the software used to generate the figures.
10. L234-237 Suggest rephrasing them and try to merge them in one sentence.
11. L246-247 Just use abbreviations here, as mentioned in sector 2.6.
12. L280-281 These elements are well shown in Figures and sector 2.3- not necessary to put them as text in the manuscript.
13. L300-301 Just use abbreviations here, as mentioned in sector 2.6.
14. L367-380 I think these two paragraphs should move to the introduction sector.
15. L437-438 Suggest merging them in one sentence.
Author Response
"Please see the attachment."

Reviewer 2 Report
This paper has extensive information about respiratory diseases and its relation to air pollution, smoking, poverty, etc. But I do not see significant modelling.
Author Response
"Please see the attachment."

Reviewer 3 Report
I suggest the authors revise the introduction of the study per the comments raised. The authors can also use the following points below as a guideline to help them come out with an interesting introduction that is more scientific.
Background & Significance: (What general background does the reader need in order to understand the manuscript and how important is it in the context of scientific research).
Problem definition: (What are the research questions to fill in the gaps of the existing knowledge body or methodology (Methods are not a contribution, but a tool to assess whether your hypothesis or predictions are supported or not supported)? I would like to see well developed arguments for predicting or proposing specific relationships in this study.
Motivations & Objectives: (Why are you conducting the study and what are the goals to achieve?)
Explain the variable selection procedure; why the authors choose these variables for the study.
I would like to suggest that authors should update the introduction, literature, and results part. Specifically, the latest research trends, and in order to highlight the academic frontier of the research, the references of the recent year need to be referenced.
The author(s) need to compare their results (Each Findings) with past studies (what was provided in the article is not compares of results but an explanation of views from past authors) and in comparing the result from the empirical investigations the author(s) should as much as possible provide a recast of the comparison made and the supposed implications or advantages of the new finding made with those discovered by past authors. This will ensure justice to the extant literature and also evincing the superiority of the current findings over the past findings.
The conclusion of the study should have a realistic empirical overview and not a summary. The conclusion should provide an overall thought from the author(s) empirical and conceptual viewpoint on why and how things exist or went the way they were discovered, what are the implications of that and what are the advantages to the key study areas under survey. Is this development attributable to the method or the variables or are these findings the reality of the situations on the ground or what is to be expected soonest etc? This should be presented in a professional, logical, and philosophical way to convey some key scientific thoughts to the readers. The present conclusion provided in the manuscript needs to be improved to a reasonable scientific standard.
What are the future research directions and limitations of this study?
In summary, the work has the potential to be published but before it should be considered for publication, it has to pass through professional proofreading and all the highlighted points above need to be corrected and implemented.
For greater clarity, the authors should cite a few articles such as:
https://doi.org/10.1007/s41651-022-00102-4
https://doi.org/10.1016/j.scitotenv.2022.154103
https://doi.org/10.1021/acs.est.1c00698
https://doi.org/10.1007/s41651-022-00108-y
Author Response
"Please see the attachment."

Reviewer 4 Report
The topic of the manuscript entitled “Patterns of Emergency Room Visits for Respiratory Diseases in New York State in Relation to Air Pollution, Poverty and Smoking” is very important and topical as it highlights how factors such as air pollution, poverty and smoking affect the rates of emergency room visits for respiratory ailments in New York State counties. Asthma, chronic obstructive pulmonary disease (COPD), lower acute respiratory disease and upper acute respiratory disease are the respiratory diseases considered for this study. To support this study, information on air pollution was obtained from the National Emissions Inventory. The paper details the results obtained showing that air pollution is the major risk factor for asthma and that the poverty factor predominates, concluding that poor people are more vulnerable to all forms of respiratory diseases. Therefore, after making the suggested changes (listed below), this article can be accepted for publication in the International Journal of Environmental Research Public Health.
Suggested modifications:
Line 13: Line 28: replace “New York State” with “New York State (NYS)”.
Line 28: replace the term (COPD) with chronic obstructive pulmonary disease (COPD).
Line 28: replace “New York State” with “New York State (NYS)”.
Line 33: replace tenth with 10th.
Line 41: replace “New York State” with “New York State (NYS)”.
Lines 66-67: perhaps it would be better to replace the acronyms PM2.5, NO2 and BC with particulate matter (PM 2.5), nitrogen dioxide (NO2) and black carbon (BC).
Line 70: perhaps it would be better to replace COPD with chronic obstructive pulmonary disease (COPD) by repeating the term in full since it is the title of the paragraph.
Line 80: replace “ER” with “emergency room (ER)”.
Line 82: perhaps it would be better to replace the acronym SO2 with sulfur dioxide (SO2).
Line 83: perhaps it would be better to replace the acronym O3 with ozone (O3).
Line 85: perhaps it would be better to replace the acronym (PM10) with particulate matter (PM10).
Line 106: perhaps it would be better to replace the acronym (CO) with carbon monoxide (CO).
Line 107: delete the hyphen at the end 1-mg/m3 by writing 1mg/m3.
Line128: replace “New York State (NYS)” with “NYS”.
Line 151: perhaps it would be better to replace “New York state” with acronym NYS.
Line 154: perhaps it would be better to replace “the below formula” with “the below Equation (1):”.
Lines 155-156: write “Population (Year) = Population 2010 + (Year - 2010)*(Population 2018 - Population 2010)/(8)” as a formula using the word insert equation function and indicate beside it (to the right of the formula) the term “(1)”.
Example:
(1)
Line 157: replace “Where” with where.
Lines 162-163: = after the term by county perhaps it would be better to insert is given “by the following Equation (2):”. Then use the word insert equation function to write the equation.
Example:
(2)
Line 170: perhaps it would be better to replace the acronym (VOCs) with “volatile organic compounds (VOCs)”.
Line 204: perhaps it would be better to replace “New York State” with acronym NYS.
Lines 261…265: Arrange the pictures in Figure 1 and their respective captions by writing them in an orderly manner.
Lines 318-319-330-331-489-490-504-505: perhaps it would be better to replace “Association” with “association”.
Line 503: perhaps it would be better to replace “Supplement” with “Supplementary Materials”.
Line 355: Delete the colon at the end of the sentence.
Lines 366-393: Delete the dot at the end of the sentence.
In the first column of the Table S2, Table 1, Table 2, Table 3, Table 4, Table 5 and Table 6 perhaps it would be better to replace “Respiratory Diseases” with “Respiratory Disease”.
Replace “chemical” with “Chemical”.
Replace “References:” with “References”.
Author Response
"Please see the attachment."

Round 2
Reviewer 3 Report
Thanks to the authors for improving their manuscript